# Study Protocol: The Evaluation Study for Social Cognition Measures in Japan (ESCoM)

**DOI:** 10.3390/jpm11070667

**Published:** 2021-07-16

**Authors:** Ryotaro Kubota, Ryo Okubo, Hisashi Akiyama, Hiroki Okano, Satoru Ikezawa, Akane Miyazaki, Atsuhito Toyomaki, Yohei Sasaki, Yuji Yamada, Takashi Uchino, Takahiro Nemoto, Tomiki Sumiyoshi, Naoki Yoshimura, Naoki Hashimoto

**Affiliations:** 1Department of Psychiatry, National Center of Neurology and Psychiatry Hospital, Tokyo 187-8551, Japan; kubotar@ncnp.go.jp (R.K.); hokano@ncnp.go.jp (H.O.); yujiyamada@ncnp.go.jp (Y.Y.); sumiyot@ncnp.go.jp (T.S.); naoyoshi@ncnp.go.jp (N.Y.); 2Department of Clinical Epidemiology, Translational Medical Center, National Center of Neurology and Psychiatry, Tokyo 187-8551, Japan; ysasaki@ncnp.go.jp; 3Department of Psychiatry, Hokkaido University Graduate School of Medicine, Sapporo 060-8638, Japan; hakiyama203@gmail.com (H.A.); a-miyazaki@med.hokudai.ac.jp (A.M.); toyomaki@gmail.com (A.T.); 4Endowed Institute for Empowering Gifted Minds, University of Tokyo Graduate School of Arts and Sciences, Tokyo 153-0041, Japan; satoru-ikezawa@g.ecc.u-tokyo.ac.jp; 5Department of Neuropsychiatry, Toho University Faculty of Medicine, Tokyo 143-8541, Japan; takashi.uchino@med.toho-u.ac.jp (T.U.); takahiro.nemoto@med.toho-u.ac.jp (T.N.); 6National Center of Neurology and Psychiatry, Department of Preventive Intervention for Psychiatric Disorders, National Institute of Mental Health, Tokyo 187-8553, Japan

**Keywords:** mental disorders, schizophrenia, developmental disorders, social cognition, social functioning, facial expression recognition, test battery, quality of life

## Abstract

In schizophrenia, social cognitive impairment is considered one of the greatest obstacles to social participation. Although numerous measures have been developed to assess social cognition, only a limited number of them have become available in Japan. We are therefore planning this evaluation study for social cognition measures in Japan (ESCoM) to confirm their psychometric characteristics and to promote research focused on social cognition. Participants in the cross-sectional observational study will be 140 patients with schizophrenia recruited from three Japanese facilities and 70 healthy individuals. In our primary analysis, we will calculate several psychometric indicators with a focus on whether they can independently predict social functioning. In secondary analyses, we will assess the reliability and validity of the Japanese translations of each measure and conduct an exploratory investigation of patient background, psychiatric symptoms, defeatist performance belief, and gut microbiota as determinants of social cognition. The protocol for this study is registered in UMIN-CTR, unique ID UMIN000043777.

## 1. Introduction

Schizophrenia is a severe mental disorder for which complete recovery is difficult to achieve by conventional therapies. As a result, patients with schizophrenia continue to have impaired social functioning. Although the prevalence of schizophrenia is a relatively low 1% [1], it is one of the top 15 causes of disability worldwide [2]. 

Social cognition is defined as “the mental operations that underlie social interactions, including perceiving, interpreting, and generating responses to the intentions, dispositions, and behaviors of others” [3]. In patients with schizophrenia or other mental disorders, social cognitive impairment is considered one of the greatest obstacles to social participation, such as interpersonal relationships, education, and employment [4].

Social cognition has become a major area of schizophrenia research, and the number of publications on this topic has increased remarkably in the past 20 years [3,5]. Measures for assessing social cognitive impairment have been developed from various perspectives. However, many of these measures are based on original theories, and reliability and validity have not been sufficiently examined for some of them [5]. To address this problem, the Social Cognition Psychometric Evaluation (SCOPE) study was conducted in the United States from 2012 to 2017. Through voting and discussion by an expert panel and two observational studies, six measures from a pool of 108 candidates were ultimately recommended for assessing social cognitive impairment in schizophrenia and other mental disorders [5,6,7]. The SCOPE study made significant advances in the assessment of social cognitive impairment, contributing a provisional test battery and serving as a foundation for future work. However, the results were based on data collected in only the United States, so the generalizability to other cultural contexts remains unclear. Notably, social cognition tasks are more sensitive than neurocognitive tasks to differences in language and culture [8]. 

In fact, there has been little effort to test the reliability and validity of the measures recommended by the SCOPE study in languages other than English or in cultural areas outside of the United States. A 2020 study of reliability and validity in Singapore by Lim et al. found that even among people who use English as a common language, reliability and validity results may differ by cultural area [9]. Going forward, as therapies are developed for social cognition, it will be absolutely paramount to examine the effects of cultural differences in the underlying assessments of social cognition. 

In Japan, some researchers have been working on developing a social cognition test battery for Japanese patients with schizophrenia [10], and validity studies have been conducted for social cognition measures such as the widely used Japanese versions of the Facial Emotion Selection Test (FEST) and Social Cognition Screening Questionnaire (SCSQ) [11,12]. However, reliability and validity have been assessed for only one of the Japanese versions of the six measures recommended by the SCOPE study. In addition, the reliability and validity of some social cognition measures currently used in Japan, including measures unique to Japan, have not been sufficiently assessed. Therefore, in order to prepare for future international clinical trials, it is necessary to assess not only the six measures recommended by the SCOPE study, but also the measures unique to Japan. 

Considering the above, we previously recruited a panel of Japanese experts on social cognition who discussed whether social cognition measures developed in the United States could be introduced in Japan, and the panel selected nine promising measures of social cognition in Japanese patients using a modified Delphi method [13]. Accordingly, the main objectives of the present study are to assess these measures selected by the domestic expert panel simultaneously in the same patients and to examine the relative merits of these social cognition measures by calculating psychometric indicators. Another objective is to hold an expert panel discussion of these psychometric indicators to produce recommendations on social cognition measures for Japanese patients.

To date, there have been studies examining cultural differences in social cognition measures between Asian and western countries [14] or attempting to establish new social cognition test batteries suited to their own specific study purposes [15,16]. However, this is the first study that will comprehensively evaluate the psychometric properties of social cognition measures in a non-English context using the framework of the SCOPE study. 

We will also examine clinical backgrounds, psychiatric symptoms, and defeatist performance belief as determinants of social cognition. Furthermore, based on the recent focus on the relation between the gut microbiome and schizophrenia [17], we will also measure gut microbiota in an exploratory investigation of social cognitive impairment, for which an effective treatment method has not yet been established. According to an experiment using germ-free mice, the gut microbiome plays an important role in cognition and social behavior [18]. When the gut microbiota of patients with schizophrenia were transplanted into germ-free mice, these animals demonstrated reduced functioning related to memory, learning, and social behavior compared with those receiving transplants of gut microbiota from healthy humans [19]. To date, five studies have been conducted on the gut microbiota in patients with schizophrenia. These studies have shown that the composition of the gut microbiota in patients is different from that in healthy individuals [20] and that gut microbiota composition is associated with symptoms of mental illness [21] and depression [22]. However, no study has yet examined the relation between social cognition and the gut microbiota in patients with schizophrenia. Therefore, in secondary analyses, we will assess the reliability and validity of the Japanese translations of the abovementioned measures and conduct an exploratory investigation of patient background, psychiatric symptoms, defeatist performance belief, and gut microbiota as determinants of social cognition. 

## 2. Methods and Analysis

### 2.1. Study Participants and Data Collection

This is a cross-sectional study. Patients with schizophrenia will be recruited from the National Center of Neurology and Psychiatry, Hokkaido University Hospital, and Toho University Omori Medical Center. Staff at each facility will screen patients based on their medical records and select eligible patients. Next, these patients will be examined by their attending physicians, who will confirm whether the patients meet the eligibility criteria. Healthy individuals will be recruited by methods such as flyers posted or distributed at the study facilities and e-mails about the study posted on mailing lists. Individuals who volunteer to participate will be interviewed by staff at the participating facilities, who will confirm the individuals’ eligibility. Individuals who meet the inclusion criteria will be given a written explanation of the study by staff at the participating facilities, who will then obtain written informed consent. The staff will assess measures with individuals who provide written consent. Anyone involved in the research agrees to participate and agrees to have details the results of the research about them published.

Of the nine tests recommended by the panel, the Penn Emotion Recognition Task (ER-40) [23] was excluded from the present study because a similar task, FEST, which measures cognitive functions related to facial expressions, is already available in Japanese. In addition, it requires the use of a unique interface and is difficult to administer compared with other measures in the same environment.

The study period for this cross-sectional study is from institutional review board (IRB) approval until 31 March 2023. Each measure will be conducted either once or twice. For healthy individuals, all assessments will be completed on the day that informed consent is obtained or within 1 week. For patients with schizophrenia, the initial assessment will be performed on the day that informed consent is obtained (day 0) or within 1 week (Table 1). In consideration of fatigue in participants, measures may be assessed over multiple days. In general, however, all measures are scheduled to be completed within several days.

### 2.2. Inclusion and Exclusion Criteria

Inclusion criteria for patients with schizophrenia are as follows: (1) primary diagnosis of schizophrenia based on the Diagnostic and Statistical Manual of Mental Disorders (DSM-5) at the time of assessment; (2) no hospitalization in the previous 2 months, no changes in prescriptions in the past 6 weeks, and no changes in prescription dosage in the past 2 weeks; (3) age 20−59 years at the time of obtaining informed consent; and (4) written informed consent to participate in the study based on an understanding of its objective and content (capacity to consent).

The inclusion criteria for healthy individuals are as follows: (1) age 20−59 years at the time of study participation; (2) confirmation of no diagnoses of mental disorders at the time of study participation; and (3) written informed consent to participate in the study.

For both patients with schizophrenia and healthy individuals, the exclusion criteria are as follows: (1) physical/mental disorders that prevent implementation of measures during study participation; (2) insufficient Japanese language ability to respond to self-reported psychological measures based on a sufficient understanding of the questions; and (3) being deemed ineligible to participate for any other reason by an attending physician or study staff.

### 2.3. Candidate Measures for Social Cognition

#### 2.3.1. Attributional Style Bias

Ambiguous Intentions and Hostility Questionnaire (AIHQ)The participant responds to questions about five situations with negative outcomes, such as questions on the cause of the situation, whether they feel the other person’s actions are intentional, and how they would respond to the situation. We obtained permission from the original authors to modify the existing Japanese version of the questionnaire by expanding the number of self-report items and removing rater-scored items, as suggested by Buck et al. [24]. The format was changed to address the limitations presented in the SCOPE study [5,6,7]. The estimated time required is 6 min [25].Intentionality Bias Task (IBT)The participant responds to 24 short sentences describing human actions within a time limit and indicates whether those actions are intentional or accidental. A Japanese version of the IBT was newly prepared for this study. The 24-question version used in the SCOPE study [7] was first translated into Japanese, and then back-translated to English for revision by the original author. In addition, modifications were made to the time conditions to account for differences in grammatical structure and average reading speed between English and Japanese. The estimated time required is 5 min [26].

#### 2.3.2. Emotion Processing

Bell Lysaker Emotion Recognition Task (BLERT)The participant views videos of actors and, using the actor’s facial expression, tone of voice, vocal timbre, and upper body movement as clues, responds to multiple-choice questions asking which emotion was being expressed. A new Japanese version, refilmed with a Japanese actor and a translated script, was prepared with permission from the original author. The estimated time required is 7 min [27].Facial Emotion Selection Test (FEST)The participant views photos of Japanese faces of different genders and ages and selects the emotion that most closely matches the expression in the photo from 7 choices (Ekman’s six basic emotions and emotionless). This test was created in Japan with reference to the Facial Emotion Identification Task (FEIT) [28]. The estimated time required is 10 min [11].

#### 2.3.3. Social Perception

Social Attribution Task-Multiple Choice (SAT-MC)The participant views animations of moving geometric figures and responds to questions about the meaning and motive of the figures’ movement. Japanese versions of both the SAT-MC I and II were newly prepared. The original English texts were translated to Japanese and then back-translated to English. The back-translations were reviewed by the original author and modifications were made to the Japanese translation as deemed necessary. The estimated time required is 10 min [29].Biological Motion Task (BM)The BM consists of two tasks that measure the ability to distinguish biological motion. In the first task, the participant distinguishes between human movement and scrambled (nonbiological) motion represented by point-lights. In the second task, scattered moving point-lights are projected onto human movement and scrambled motion, and the participant is asked to distinguish between them. The number of scattered moving point-lights varies depending on the movement in each trial. The estimated time required is 10 min [30].

#### 2.3.4. Theory of Mind

Hinting Task (Hinting)The participant reads and hears a dialogue between two characters and identifies the true intention behind one character’s indirect speech. A new Japanese version was prepared with permission from the original author. In this new version, in order for the participant to take the test by themself on a computer, they answer each question twice, first without a hint, and then with a hint. The participant answers the questions aloud and these answers are recorded. Based on the recording, 2 points are given for correct answers in the no-hint phase, 1 point for correct answers in the hint phase, and 0 points for not answering any of the questions correctly. The estimated time required is 6 min [31].Metaphor and Sarcasm Scenario Test (MSST)The participant reads 10 passages involving metaphors and sarcasm and chooses the answer that most accurately describes the passage. The estimated time required is 8 min [32].

### 2.4. Other Measurements

Background informationInformation such as sex, age, educational background, medical history, treatment history, primary disease, history of allergies, and history of side effects will be collected from medical records or interviews with participants.Mini-International Neuropsychiatric Interview (M.I.N.I.)The M.I.N.I is a structured interview designed to diagnose mental disorders. The present study uses the version of the M.I.N.I adapted to the DSM-V. The M.I.N.I will be conducted after obtaining informed consent to confirm history of mental disorders in healthy participants. For participants with schizophrenia, the M.I.N.I. will be performed after obtaining consent to determine whether they meet the inclusion criteria. The interview will take roughly 30 min [33,34].Japanese Adult Reading Test-25 (JART-25)The JART-25, a measure of verbal IQ, consists of 25 two-character kanji compound words that participants are asked to read aloud. In patients with schizophrenia, the JART-25 is considered to reflect premorbid verbal IQ. The estimated time required is 5 min [35].Positive and Negative Syndrome Scale (PANSS)The PANSS assesses overall psychiatric symptoms in schizophrenia via interview. The scale is composed of 30 items: 7 items on positive symptoms, 7 items on negative symptoms, and 16 items on general psychopathology symptoms. This study will use the Japanese version of the PANSS, which was translated by the Japan Young Psychiatrists Organization. The estimated time required is 30 min. Assessment by an informant (estimated time required: 10 min) is also performed [36,37].Brief Negative Symptom Scale (BNSS)The BNSS is a 13-item scale that assesses negative symptoms in schizophrenia via interview. The estimated time required is 15 min [38,39].Defeatist Performance Beliefs (DPB) ScaleThe DPB Scale is a self-administered scale for assessing negative beliefs about oneself. The estimated time required is 5 min [40,41].General Causality Orientations Scale (GCOS)The GCOS is a self-administered scale that assesses individual tendencies regarding three different motivational orientations (autonomy, control, and impersonal, which correspond to intrinsic motivation, extrinsic motivation, and amotivation, respectively). The estimated time required is 10 min [42].Self-Assessment of Social Cognition Impairments (ACSo)The ACSo is a 12-item self-administered questionnaire that examines subjective complaints regarding four different domains of social cognitive impairment. The estimated time required is 5 min [43,44]. The original French text was translated to Japanese and then back-translated to French. The back-translation was reviewed by the original authors and modifications were made to the Japanese translation as deemed necessary.Observable Social Cognition Rating Scale (OSCARS)The OSCARS is an 8-item scale that comprehensively examines subjective complaints regarding social cognitive impairment. The scale involves a self-report and an objective assessment from an informant close to the participant. The estimated time required is 5 min [45,46]. The original English text was translated to Japanese and then back-translated to English. The back-translations were reviewed by the original authors and modifications were made to the Japanese translation as deemed necessary.Brief Assessment of Cognition for Schizophrenia, Japanese Version (BACS-J)The BACS, which is a standardized test battery for which validity has been examined, was developed to assess cognitive impairment in schizophrenia. The assessment, which is currently widely used for psychiatric disorders, consists of verbal memory, working memory, motor speed, attention, verbal fluency, and executive functions. The estimated time required is 30 min [47,48].University of California, San Diego Performance-based Skill Assessment—Brief (UPSA-B)The UPSA-B measures functioning in two domains: financial management and communication (by telephone), in a role-playing scenario that models daily living. The estimated time required is 10 min [49,50].Specific Levels of Functioning Scale (SLOF)The SLOF objectively assesses social functioning by integrating the results of interviews with the participant and a close informant with results from a self-administered questionnaire. The estimated time required is 10 min [51,52].Gut microbiotaUsing a specialized gut microbiota measurement kit, we will measure gut microbiota as described previously [53,54]. Based on reference sequences, we will categorize bacteria into operational taxonomic units and calculate the occupancy rate (the percentage of the gut microbiome occupied by a given bacterium) of each bacterium at the genus level.

### 2.5. Sample Size Calculation

The present study aims to simultaneously assess measures of social cognition and quantitatively determine which measures are psychometrically superior. We will calculate various psychometric indicators with an emphasis on whether they can independently predict social functioning; specifically, we will emphasize the incremental validity of social functioning (defined as a significant increase in the coefficient of determination when social functioning testing is added after neurocognitive function testing has been included in advance as an independent variable) as determined by hierarchical multiple regression analysis. Sample size was calculated based on hierarchical multiple regression analysis.

In multiple regression analysis, f^2^ indicates effect size. According to Cohen [55], f^2^ values of 0.02, 0.15, and 0.35 represent small, moderate, and large effect sizes, respectively. In the present study, Model 1 uses the six cognitive domains assessed by the BACS-J (verbal memory, working memory, motor speed, attention, verbal fluency, and executive function) as explanatory variables and social functioning (SLOF and UPSA) as the response variable. In Model 2, each of the social cognition measures is added one at a time. Based on a previous study [7], the effect size of Model 1 is assumed to be 0.25, which is halfway between moderate and large. The present study requires a sample size sufficient to give an increase in the effect size of Model 2 to large (0.35), that is, to detect social cognition measures that, when added, boost the predictive ability of social functioning from halfway between moderate and large. The sample size necessary for detection with α = 0.05 and β = 0.2 is 99 participants (SAS 9.4). To conduct this analysis, all measures must be completed. In view of the high dropout rate of 40% in a previous study [7], we assumed a dropout rate (dropping out at any time before the final analysis) of 30% in the present study; therefore, we set the target sample size at 140 participants.

The purpose of recruiting healthy individuals for this study is to estimate reference values for social cognition measures in the general population in an exploratory fashion. We will recruit 8 participants in each of eight classifications (men and women in their 20s, 30s, 40s, and 50s). In consideration of dropouts, we set the target sample size at 70 participants.

### 2.6. Statistical Methods

For participants’ background information, we will calculate summary statistics separately for the patient group and the healthy group and compare them between groups. To examine the relative merits of social cognition measures, we will calculate the following psychometric indicators. For reliability, in addition to performing correlation analysis and assessing test–retest reliability, we will calculate Cronbach’s alpha to assess internal consistency. For changes from the initial test to the retest (learning effect), we will calculate Cohen’s d and state the respective frequencies of floor/ceiling effects. In addition to performing simple correlation analysis for each social cognition test and social functioning, we will perform multiple regression analysis to determine the overall extent to which each of the social cognition measures predicts social functioning. For each social cognition measure, we will also examine the incremental validity of social functioning (defined as a significant increase in the coefficient of determination when social functioning testing is added after neurocognitive function testing has been included in advance as an independent variable) as determined by hierarchical multiple regression analysis. Furthermore, we will calculate the means and standard deviations (SDs) for the time required for each social cognition measure and participants’ subjective assessments of each measure separately for the patient group and the healthy group. We will also calculate means and SDs of results for each social cognition measure separately for the patient group and the healthy group, compare means between the two groups, and note Cohen’s d.

The details of the above statistical analysis will be shown separately in a statistical analysis protocol that will be drafted by the time final data entry is closed. Interim analysis will not be conducted. For missing data, we are considering listwise deletion as the first option. However, if listwise deletion results in a high percentage of missing data, we will consider statistical analysis that imputes missing data.

## 3. Ethics and Data Management

### 3.1. Ethical Considerations

The written explanation and consent form, which have been approved by an IRB, will be handed to participants. Following thorough written and oral explanations, we will obtain informed consent from participants based on their own free will. In the event that information is obtained or changes to the study protocol are made that might affect a participant’s consent, we will provide this information to the participant promptly. We will then confirm the participant’s willingness to participate in the study, revise the explanation form and consent form with IRB approval, and obtain the participant’s informed consent once again. The protocol for this study is registered in the UMIN Clinical Trials Registry (UMIN-CTR), unique ID UMIN000043777.

In terms of possible harm, this is a cross-sectional observational study that does not involve any intervention and is therefore presumed not to inflict any burden related to invasiveness. However, participants will be burdened financially by transportation expenses. The measures take 6 h to complete, meaning that we cannot rule out the possibility of fatigue. In the event of fatigue or any other situation that may make continued testing inappropriate, we will include breaks, readjust the testing schedule, or take any other measures necessary to improve the situation. In addition, the principal investigator or co-investigators will promptly conduct appropriate examination and treatment.

### 3.2. Patient and Public Involvement

Ken Udagawa of the Community Mental Health & Welfare Bonding Organization (COMHBO) participated in the research team and joined study group meetings. He gave us various forms of advice, such as revising expressions in measures and questionnaires to make them easier for patients with schizophrenia to understand. In addition, Daisuke Haga of One More (managed by the Japan Learning Association) performed assessment of the social cognition measures with 5 patients with schizophrenia in his organization and gave us information regarding the measures’ usability.

### 3.3. Data Management and Monitoring and Auditing

Participants’ personal information will be managed by a personal information manager. All samples and information collected during the study will be anonymized at the time of collection by an anonymization manager to prevent identification of participants by sample or information. We will anonymize information by assigning a number unique to the study to each participant’s consent form (which contains personally identifiable information such as name and address) and test results and then create a correspondence table. We will store original copies of test forms and data after deleting information that can be used to identify individual participants (name, address, date of birth, etc.). All paper materials relating to individuals will be stored in locked document storage cabinets in the facilities where the research is being conducted (National Center of Neuropsychiatry: locked cabinet in the Translational Medical Center; Hokkaido University: locked cabinet in the Department of Psychiatry; Toho University: locked cabinet in the Department of Neuropsychiatry). The key will be kept by the personal data manager, and the area where personal data are handled will be within the relevant facility, and the data will always be stored in a locked cabinet after use.

Electronic data will be stored using flash memory devices that require mandatory encryption and password authentication. Only the principal investigators at each institution (National Center of Neurology and Psychiatry: Ryo Okubo, Hokkaido University: Naoki Hashimoto, Toho University: Takahiro Nemoto) will have access to the stored flash memory. Access may be granted temporarily to persons approved by the principal investigator as necessary to carry out the research (e.g., data entry and analysis). Measurement of gut microbiota will be outsourced to a testing company called Cykinso, where samples will be managed with numbers unique to the study assigned by the study secretariat. Thorough caution will be taken so that names and other personally identifiable information will not be given to Cykinso. DNA extract from stool-derived gut bacteria will be disposed of after analysis is completed. 

Because the present study has no intervention, we will not conduct monitoring or auditing.

## 4. Discussion

### 4.1. Dissemination: Process for Final Recommendation

With reference to the results of this study, we will assign all indicators one of three grades based on criteria that were determined in advance through a vote by an expert panel. The grades are “appropriate,” “appropriate with reservation,” and “use with caution.” The appropriateness of each measure depends on its purpose of use, such as for clinical research (observational studies and interventional studies) or for only clinical purposes such as screening and rehabilitation assessment. Therefore, when recommendations are finalized, we will grade the measures according to their intended use and list advantages and points of caution for each. 

The members of the expert panel will grade each measure based on the assessment criteria determined by an expert panel in 2020 and with reference to the results of a validity study conducted in 2021 for psychometric assessments. The panel members will vote on which grade best applies to each measure, with additional rounds of discussion and voting conducted until a consensus of ≥80% (at least 8 of 9 members) is reached. The number of members who object to the final vote and their reasons for objecting will be recorded and appended to the grading results when they are published.

Good practicality is defined as each social cognition domain taking less than 15 min to administer, while good tolerability is defined as the test having a small burden based on subjective assessment by the participants. For test–retest reliability, a correlation coefficient of ≥0.6 is defined as “appropriate.” For utility as a repeated measure, we will place importance on the absence of a floor effect in both the first and second measurements. However, in order to use measures as outcomes in interventional studies, we will place importance on the absence of both a floor effect and a ceiling effect in both the first and second measurements.

For validity, we will place importance on a pronounced difference between patients with schizophrenia and healthy individuals and on a strong correlation with social functioning. We will also examine incremental validity, that is, a further increase in the power to predict social functioning resulting from the addition of social cognition performance to neurocognitive performance. For the sake of international comparisons, we will prioritize the six measures recommended by the SCOPE study (BLERT, Hinting, ER-40, and IBT) conducted in the United States when they are equally applicable in Japanese clinical practice.

At a point between the completion of the data analysis and the conclusion of the research project, the results will be linked to anonymized data and presented at relevant academic conferences and in academic journals without revealing identification of participants.

### 4.2. Prospects for the Future

Generating recommendations for standard measures of social cognition for Japanese patients may lead to (1) the promotion of education/employment support tailored to social cognitive impairment, and (2) greater participation by Japanese researchers in international collaborations on social cognitive impairment and identification of factors in mental disorders in general which worsen social function.

Furthermore, through subsequent research using these social cognition measures, we aim to (1) determine the needs of not only patients with social cognitive impairment but also healthcare workers, (2) identify social cognitive impairments that prominently affect forms of social participation such as employment and education, and (3) support the development of intervention programs to treat social cognitive impairment effectively.

We also hope that our research will help to identify differences in social cognition between countries and ethnic groups, and inform future efforts to examine social cognitive impairment in patients with schizophrenia across populations with various biological and cultural characteristics.

## Figures and Tables

**Table 1 jpm-11-00667-t001:** Testing schedule.

	Day 0–7	Day 0–7 (All Measures Below to Be Completed within 2 Days)	Day 14–42
Informed consent (patients, healthy individuals)	○		
Background information (patients, healthy individuals)	○		
M.I.N.I. (patients, healthy individuals)	○		
JART-25 (patients, healthy individuals)	○		
PANSS (patients)		○	
BNSS (patients)		○	
DPB (patients *)		○	
BACS-J (patients *)		○	
UPSA-B (patients *)		○	
SLOF (patients *)		○	
Japanese versions of social cognition measures (patients *, healthy individuals)		○	○
Scale of subjective difficulty of social cognition measures (patients *, healthy individuals)		○	
Gut microbiota (patients, healthy individuals)	□		
Duration (min)	40	180 (Healthy individuals: 80)	70

○: mandatory, □: optional. * Measures with an asterisk next to “patients” are conducted with patients for the purposes of this study, while measures without an asterisk are conducted as part of standard clinical practice. BACS-J, Brief Assessment of Cognition for Schizophrenia, Japanese Version; BNSS, Brief Negative Symptom Scale; DPB, Defeatist Performance Beliefs; JART-25, Japanese Adult Reading Test-25; M.I.N.I., Mini-International Neuropsychiatric Interview; PANSS, Positive and Negative Syndrome Scale; SLOF, Specific Levels of Functioning Scale; UPSA-B, University of California, San Diego Performance-based Skill Assessment—Brief.

## Data Availability

Not applicable.

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
