# Peer review of "Study Protocol: The Evaluation Study for Social Cognition Measures in Japan (ESCoM)"

_jpm, 2021, doi:10.3390/jpm11070667_

Round 1

Reviewer 1 Report

Review of the Report:

The objective of this study was to assess and examine the measures of social cognition and evaluating psychometric indicators applicable to Japanese patients. 

It is an interesting proposal to assess and measure the gut microbiom in schizophrenic patients in relationship with social cognition. The gut microbiom has been an importance and innovative method of assessing myriad of health issues and social behaviors. 

It's a very well written report with suitable methodology and lengthy descriptions of the instruments to be administered to the cohort. 

Would be great to see the results after the study has been conducted.

Author Response

The objective of this study was to assess and examine the measures of social cognition and evaluating psychometric indicators applicable to Japanese patients.

It is an interesting proposal to assess and measure the gut microbiom in schizophrenic patients in relationship with social cognition. The gut microbiom has been an importance and innovative method of assessing myriad of health issues and social behaviors.

It's a very well written report with suitable methodology and lengthy descriptions of the instruments to be administered to the cohort.

Would be great to see the results after the study has been conducted.

Reply: Thank you so much for your positive feedback.

Reviewer 2 Report

The authors described a study protocol aimed to evaluate social cognitive measurements in Japan. While the manuscript is well-written, the significance of the current protocol is unclear.

It is important to address the importance of the protocol proposed in the manuscript, as well as compare it with existing protocols. It is unclear why the proposed protocol differs from others or even needed.

Also, it would be interesting if the authors can discuss whether or how this protocol can be generalized to countries other than Japan considering the biological and cultural variabilities. 

Author Response

The authors described a study protocol aimed to evaluate social cognitive measurements in Japan. While the manuscript is well-written, the significance of the current protocol is unclear.

Reply: Thank you for your positive feedback.

  1. It is important to address the importance of the protocol proposed in the manuscript, as well as compare it with existing protocols. It is unclear why the proposed protocol differs from others or even needed.

Reply: Thank you for your important comment. In accordance with your comment, we have revised the introduction section in our manuscript as follows:

(Page2, line91- Page3, line100)

To date, there have been studies examining cultural differences in social cognition measures between Asian and western countries[14], or attempting to establish new social cognition test batteries suited to their own specific study purposes [15,16]. However, this is the first study that will comprehensively evaluate the psychometric properties of social cognition measures in a non-English context using the framework of the SCOPE study.

We added the related reference as follows:

Lee, H.S.; Corbera, S.; Poltorak, A.; Park, K.; Assaf, M.; Bell, M.D.; Wexler, B.E.; Cho, Y.I.; Jung, S.; Brocke, S.; et al. Measuring theory of mind in schizophrenia research: Cross-cultural validation. Schizophrenia research 2018, 201, 187-195, doi:10.1016/j.schres.2018.06.022.

Social Cognitive Assessment in Autism and Schizophrenia ((ClaCoS)). Available online: https://www.clinicaltrials.gov/ct2/show/NCT02660775 (accessed on 6 July 2021).

Peyroux, E.; Prost, Z.; Danset-Alexandre, C.; Brenugat-Herne, L.; Carteau-Martin, I.; Gaudelus, B.; Jantac, C.; Attali, D.; Amado, I.; Graux, J.; et al. From "under" to "over" social cognition in schizophrenia: Is there distinct profiles of impairments according to negative and positive symptoms? Schizophr Res Cogn 2019, 15, 21-29, doi:10.1016/j.scog.2018.10.001.

  1. Also, it would be interesting if the authors can discuss whether or how this protocol can be generalized to countries other than Japan considering the biological and cultural variabilities.

Reply: Thank you for your important comment. In accordance with your comment, we have revised the discussion section in our manuscript as follows:

(Page10, line465- Page10, line469)

We also hope that our research will help to identify differences in social cognition between countries and ethnic groups, and inform future efforts to examine social cognitive impairment in patients with schizophrenia across populations with various biological and cultural characteristics.

In addition, there was one place where the citation was incorrect, and I corrected it as follows:

(Page1, line43- Page2, line47)

the mental operations that underlie social interactions, including perceiving, interpreting, and generating responses to the intentions, dispositions, and behaviors of others